# FoSAM: Focus-Oriented Adaptive Token Sampling for Efficient Segment Anything in Augmented Reality

## Abstract

Augmented Reality (AR) encompasses transformative technologies that are redefining how humans interact with their environment. A key component of AR is image segmentation, which breaks down the user's front-view scene into distinct regions for analysis. This process is essential for accurately overlaying digital content onto the physical world by detecting and isolating relevant objects. However, despite its importance, image segmentation poses significant computational demands and latency issues on AR devices, which can severely impact the overall user experience. In this paper, we propose *Focus-Oriented Segment Anything Model* (FoSAM), a framework built upon the Segment Anything Model (SAM) that utilizes real-time gaze data to focus segmentation on regions of interest, substantially lowering computational cost. Experimental results show that FoSAM reduces computational cost by over $50\times$, enabling a seamless visual experience for users, as confirmed by our real-world user study. The code is provided at `https://anonymous.4open.science/r/FoSAM-D627`.

## 1 Introduction

Image segmentation Long et al. (2015); Badrinarayanan et al. (2017); Kirillov et al. (2023); Xie et al. (2021) is a core task in computer vision that involves dividing an image into meaningful regions to support visual content analysis and interpretation. Building on this, instance segmentation He et al. (2017a); Bolya et al. (2019); Wang et al. (2020); Yang et al. (2019) identifies and outlines each individual object within a scene. This capability is particularly crucial in augmented reality, where accurate object detection and separation enable precise interaction and seamless overlay of virtual content onto the physical environment, enhancing both immersion and contextual awareness.

Instance segmentation serves as a foundational component for numerous AR applications. For example, in educational contexts (Figure 1 (a)), segmentation can detect individual components of complex diagrams the user is viewing, enhancing student engagement and understanding. These segmented objects can also be passed to downstream applications (e.g., vision-language models (VLMs)), to provide detailed explanations or context-aware information. In an AR-assisted grocery shopping scenario, real-time segmentation enables users to identify products on a shelf as they look at them. Furthermore, segmentation enables direct object manipulation, allowing users to edit or interact with specific elements in their environment, creating a seamless bridge between the physical and virtual worlds. Additional AR use cases leveraging image segmentation are discussed in González Izard et al. (2019; 2020); Tanzi et al. (2021); Alhaija et al. (2017).

The Segment Anything Model (SAM) Kirillov et al. (2023); Ravi et al. (2024) is among the most advanced models for image segmentation today. However, despite its strong performance, SAM is computationally intensive, making it difficult to deploy on resource-limited AR devices that handle high-resolution imagery. This results in high overhead and latency, degrading overall system performance and user experience.

Unlike typical image processing scenarios, AR device users display distinct behavioral patterns. They tend to focus on small, specific areas within a scene before shifting their attention elsewhere. For instance, as shown in Figure 1 (b), a user wearing AR glasses may first focus on the door (left) and then turn their head to look at the bookshelf (right). This behavior naturally segments the video

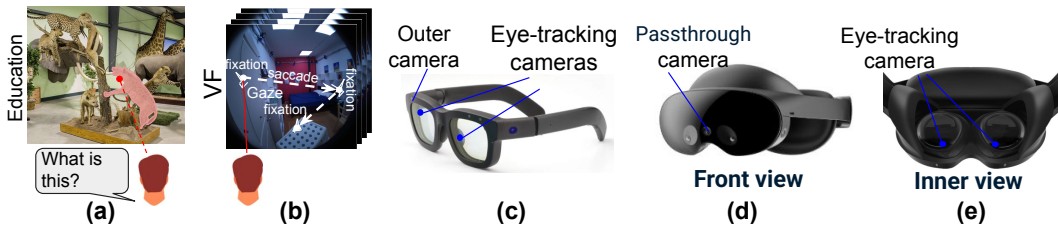

Figure 1: (a) Application of VLM in AR. (b) An example on gaze movement for the AR scenario. (c) Meta Orion AR glass. (d) Front view and (e) innter view of Meta Quest Pro VR headset.

stream into distinct video fragments (VF) based on head movements. In the first VF, where the user's gaze remains on the door across several frames, the segmentation results can be reused, avoiding redundant computation. Similarly, in the second VF, segmentation efforts can be focused solely on the bookshelf. Moreover, as illustrated in Figure 1 (a), it is often beneficial to restrict segmentation to the instances of interest (IOI) currently under the user's gaze, as this typically reflects the user's active focus. This leads to a more efficient approach for AR image segmentation: **prioritize processing for gaze-identified IOIs while ignoring non-essential regions**. This strategy aligns with the principles of *foveated rendering* Patney et al. (2016), which improves rendering efficiency by showing full-resolution detail only within the user's line of sight, reducing visual fidelity in peripheral areas to save computations.

In this work, we aim to reduce the high computational cost of segmentation in AR by leveraging natural human eye movement. We introduce a novel segmentation framework based on Efficient SAM Xiong et al. (2024) (ESAM), a lightweight version of the original SAM that maintains nearly the same performance. By limiting segmentation to only the regions the user is actively viewing, our proposed variant, called *focus-oriented SAM* (FoSAM), achieves more than a $50\times$ reduction in processing latency without compromising the user's visual experience. Our contributions are:

- We introduce a novel and important perspective on instance segmentation with SAM that exploits human eye behavior to reduce computational costs in AR settings. A simple demo can be found in `https://anonymous.4open.science/w/FoSAM-D627`).

- We propose *FoSAM*, a lightweight segmentation framework built on ESAM, which processes high-resolution input images and performs instance segmentation on the IOI with extremely low computational cost.

- Building on FoSAM, we introduce *FoSAM Streaming Algorithm* (FSA), an efficient instance segmentation framework tailored for real-time AR/VR applications. FSA, exploits temporal continuity across frames and human gaze patterns to optimize segmentation, delivering enhanced performance in dynamic AR environments.

## 2 BACKGROUND AND RELATED WORK

### 2.1 LITERATURE REVIEW ON SEGMENTATION

Semantic segmentation Shelhamer et al. (2014); Badrinarayanan et al. (2015); Chen et al. (2017); Touvron et al. (2020); Cheng et al. (2021); Minaee et al. (2020); Li et al. (2023); Xiong et al. (2019); Cheng et al. (2019) is a key task in computer vision that involves partitioning an image into distinct regions or segments to simplify content analysis and interpretation. A more advanced form of this task is instance segmentation He et al. (2017b); Li et al. (2020); Neven et al. (2019); Brabandere et al. (2017), which aims to distinguish individual instances of the same object class. Unlike semantic segmentation, which labels each pixel without differentiating between instances of the same object class, instance segmentation provides a finer level of detail by distinguishing each instance. To improve segmentation efficiency, previous research has focused on developing learnable input downsampling techniques that adjust sampling resolution in a selective manner. In Recasens et al. (2018), the authors propose a saliency-based distortion layer for convolutional neural networks that enhances spatial sampling of input data in image classification tasks. Subsequent works, such as Jin et al. (2021); Thavamani et al. (2021); Marin et al. (2019); Zeng et al. (2025), follow similar approaches by learning a saliency score for each pixel to guide the downsampling process.

Previous approaches perform image downsampling at the pixel level using Convolutional Neural Networks (CNNs). In contrast, FoSAM is a transformer-based model that adaptively selects tokens of input image at token level, guided by the user's gaze location and the shape of the IOI. Prior work Marin et al. (2023) shows that performing naive downsampling on the token map severely degrades performance. Therefore, porting pixel level downsampling methods to the token level is not feasible. Our FoSAM is fundamentally different from these approaches. Furthermore, previous studies Yang et al. (2019); Wang et al. (2021); Yan et al. (2023); Rajič et al. (2023); Lin et al. (2021); Wu et al. (2022) on video instance segmentation process consecutive frames together. This methods leverage temporal correlations across frames to improve performance; however, it introduces considerable latency, as processing can only begin once all frames are available.

## 2.2 HUMAN EYE BEHAVIOR IN AR/VR ENVIRONMENT

The human eye operates in three main modes of movement: fixation, where the eye remains stationary and focuses on a single point; saccadic movements, rapid, jerky shifts in gaze from one target to another; and smooth pursuit, where the eye follows a moving object in a smooth manner. During a saccade, the visual system's sensitivity is reduced. This decrease in sensitivity helps prevent the brain from perceiving the blur caused by the swift movement of the eyes.

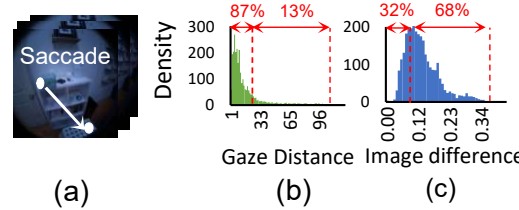

Figure 2: (a) Video Fragments in the user's view. (b) Distribution of the gaze distances. (c) Distribution of the image differences.

Figure 1 (c-e) present the Meta Aria AR glasses Engel et al. (2023) and Meta Quest Pro Inc. (2022), respectively. Both devices feature an outward-facing camera that continuously captures high-resolution images of the user's front view, along with an inward-facing eye-tracking (ET) camera that records monochrome images of the user's eyes. Most modern AR devices are equipped with eye trackers that takes the eye images and provide highly precise gaze direction estimates with minimal latency. To investigate human gaze behavior while using AR devices, we perform an in-depth analysis using the Aria Everyday Activities Dataset Lv et al. (2024). This dataset contains 143 sequences of frames captured by the AR device.

As the user adjusts their head orientation, the front view also changes. To quantify these head movements, we calculate the image difference by measuring the Euclidean distance between corresponding pixels of consecutive frames. If the pixel difference is below a certain threshold, the frames are considered highly similar and nearly indistinguishable to the human eye. These frames are grouped into a single video fragment (VF), as shown in Figure 1 (b) and Figure 2 (a). As shown in Figure 2 (c), 32% of consecutive frames exhibit less than 9% pixel value changes, **suggesting a similarity between the consecutive frames and potential for reusing segmentation results across frames**.

Furthermore, within each VF, segmentation results can be reused if the gaze remains relatively stable and consistently points to the same IOI. To support this, we analyze the distances (in pixels) between consecutive gaze locations within a VF. Our analysis reveals that a threshold of 22 pixels effectively groups gaze locations during the fixation phase, where the user focuses on a single IOI. Gaze distances exceeding this threshold indicate a saccade, as seen with the rapid gaze changes in VF2 of Figure 2 (a). As shown in Figure 2 (b), 87% of the frames within each VF have a gaze distance of less than 22 pixels, indicating that AR users typically focus on one or two IOIs during each VF. This provides an opportunity to enhance image segmentation efficiency by **focusing processing on the IOI and reusing segmentation results when gaze shifts are minimal**.

## 2.3 SEGMENTATION TASK LATENCY

Despite its significance, the segmentation task presents considerable computational challenges, especially on resource-constrained AR devices, primarily because of the high resolution of images these devices capture. This heavy data load leads to significant computational latency, severely limiting performance and responsiveness. To explore this issue, we evaluate the processing latency of the

Table 1: Processing latencies of segmentation models on AR platform.

| Image Size | SAM-B | ESAM-S | ESAM-T |
|---|---|---|---|
| $1024^2$ | 1897.4 ms | 600.3 ms | 279.7 ms |
| $640^2$ | 441.8 ms | 102.6 ms | 46.7 ms |

image segmentation task using well-established neural networks, including Segment Anything Base model (SAM-B) Kirillov et al. (2023); Ravi et al. (2024), Efficient SAM Tiny (ESAM-T) Xiong et al. (2024), and ESAM Small (ESAM-S). The latencies are profiled on the Qualcomm XR2 Gen2 platform Qualcomm Technologies, Inc. (2024b) using the Qualcomm AI Hub toolkit Qualcomm Technologies, Inc. (2024a), which is equipped by Meta Quest Pro Inc. (2022) and Meta Aria AR glasses Engel et al. (2023). Table 1 shows that the processing latencies for $1024 \times 1024$ input images are 1897.4ms, 600.3ms, and 279.7ms for SAM-base (SAM-B), Efficient SAM-small (ESAM-S), and Efficient SAM-tiny (ESAM-T), respectively. Furthermore, even with a smaller input resolution of $640 \times 640$, the latency still ranges from 441.8ms to 46.7ms. These latency fall short of the threshold required for a seamless visual experience, as prior studies indicate that latencies below 30 ms are essential for maintaining optimal visual fluidity Kaaresoja et al. (2014); vis; Albert et al. (2017).

## 3 METHODOLOGY

Figure 3 illustrates the computational pipeline of the FoSAM framework. During runtime, the inward-facing sensor of the AR/VR device continuously captures images of the user's eye and sends them to the gaze tracker, which estimates the gaze direction instantaneously with high accuracy (5 to 10 milliseconds) Stein et al. (2021); Hou et al. (2024). At timestep $t$, this estimated gaze direction $g_t$, along with a high-resolution image $I_t$ captured by the front-facing camera, is provided as input to FoSAM. FoSAM then produces a segmentation map $M_t$ focused on

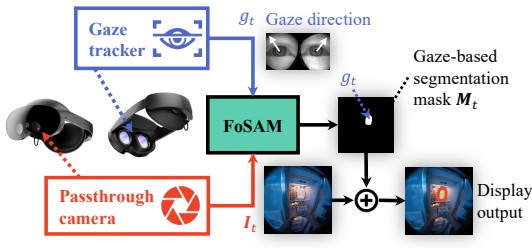

Figure 3: System Deployment of FoSAM.

the IOI, which can be reused across frames where the gaze location remains similar. To simplify notation, we omit the subscript $t$ in the following sections. The core idea of FoSAM is to employ a lightweight token selector to identify and select a subset of image tokens, which are then passed to the segmentation network. By processing only the selected tokens, the segmentation network significantly reduces its computational overhead.

### 3.1 PRELIMINARIES

FoSAM builds on ESAM Xiong et al. (2024), which includes three main components: an *image encoder* $E(.)$, a *prompt encoder* $R(.)$, and a *mask decoder* $D(.)$. Given an input RGB image $I \in \mathbb{R}^{H \times W \times 3}$, it is first tokenized and positionally embedded into a set of tokens $X = \{x_{ij}\}$ with a size of $H_e \times W_e$, where each token $x_{ij} \in \mathbb{R}^{1 \times d_t}$. $H_e$ and $W_e$ equal $\frac{H}{P}$ and $\frac{W}{P}$, where P is the patch size of the tokenizer. These tokens are passed through the image encoder to produce $Y = \{y_{ij}\} = E(X)$, which retains the same spatial resolution. In parallel, a point-based gaze prompt $g$ is processed by the prompt encoder $R(.)$, and its output is fused with $Y$ in the decoder to generate the final segmentation mask $M \in \mathbb{R}^{H \times W}$. In ESAM, the image encoder $E(.)$ is a 12-layer Vision Transformer, and the decoder $D(.)$ consists of a pre-trained two-layer CNN and a two-layer ViT. Since $E(.)$ dominates the computational cost, we focus on optimizing it by introducing a *gaze-guided Token Importance Encoder (TIEncoder)* $S(.)$, a *Gaussian Predictor* $G(.)$, and an *Adaptive Sampling module*. Together, they will selects a subset of $K$ important tokens from $X$ based on the gaze direction $g$, reducing the input size to $E(.)$ from $H_e \times W_e$ to $K$ and greatly decrease the computational cost of the segmentation task. The entire selection process is end-to-end trainable, resulting in strong accuracy performance. Next is the design of our TIEncoder $S(.)$.

### 3.2 FOSAM FRAMEWORK

#### 3.2.1 GAZE-GUIDED TOKEN IMPORTANCE ENCODER

The gaze-guided TIEncoder is designed to assign a Token Importance Score (TIScore) to each token $X$ based on the user's gaze direction $g$. A common approach predicts token-wise importance scores independently for each token Liu et al. (2021; 2023); Kockwelp et al. (2025); Rao et al. (2021); Tang et al. (2022), and then removes those with low scores. However, this strategy often leads to

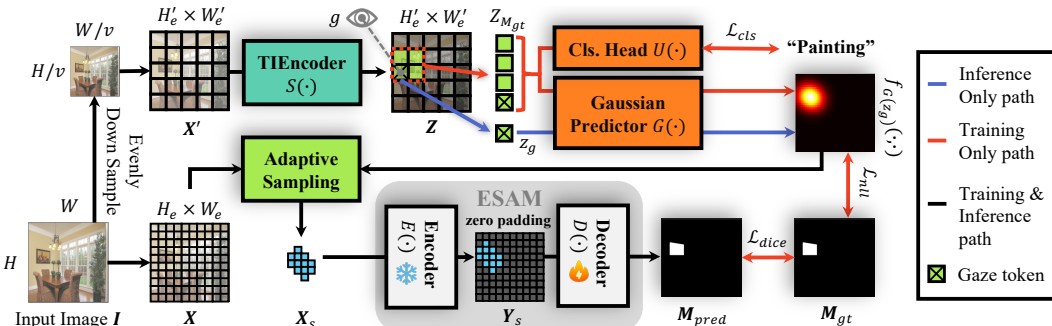

Figure 4: An overview of FoSAM framework.

poor accuracy because it lacks spatial coherence. The independently predicted scores can result in fragmented TIScore map with missing regions, which negatively affect the segmentation quality of the image decoder. Additionally, this approach requires computing a TIScore for every token, resulting in computational complexity that grows linearly with the number of tokens. This significantly limits the potential for reducing model size and improving efficiency.

To address these limitations, we propose estimating the TIScores of tokens using a 2D Gaussian distribution. The top portion of Figure 4 illustrates the overall pipeline, with the black and blue arrows show the inference flow. The input image $I$ is first uniformly downsampled from $H \times W$ to $\frac{H}{v} \times \frac{W}{v}$, resulting in a lower-resolution image $I'$, where $v$ denotes the downsampling ratio. This step effectively reduces the computational cost of the TIEncoder function $S(.)$. Next, $I'$ is passed through a pre-trained ESAM tokenizer and a positional embedding module to generate token features $X' = \{x'_{ij}\}$ with dimensions $H'_e \times W'_e$. $X'$ is then processed by the TIEncoder $S(.)$, which outputs $Z = S(X')$ containing the same number of tokens. Tokens $Z$ are selected based on their spatial proximity to the gaze point $g$ obtained from the gaze tracker. The token nearest to $g$, denoted as $z_g \in Z$, captures essential information about the IOI, such as its shape and scale, and serves as the foundation for predicting a bounding window that encloses the IOI.

To simplify the window generation process while preserving spatial continuity, we introduce a Gaussian predictor to estimate the TIScore of each token $x \in X$. This predictor, $G(\cdot)$, takes $z_g$ as input and outputs the parameters of a 2D Gaussian distribution: the mean $(\mu_x, \mu_y)$, standard deviations $(\sigma_x, \sigma_y)$, and correlation coefficient $\rho$. Let $f_{G(z_g)}(.,.)$ represent the probability density function of Gaussian distribution with the parameter specified by $G(z_g)$. The TIScore for each token $x \in X$ is then computed as its likelihood under the predicted 2D Gaussian distribution, denoted as $f_{G(z_g)}(c_i(x), c_j(x))$, where $c_i(x)$ and $c_j(x)$ is the 2D coordinate of $x$ within the token map.

### 3.2.2 ADAPTIVE TOKEN SAMPLING MECHANISM

Based on the TIScores generated by the 2D Gaussian map, only a subset of tokens with the highest scores are retained. Specifically, a larger IOI results in more tokens receiving high TIScores, corresponding to a Gaussian distribution with lower variance. Conversely, smaller IOIs produce higher-variance distributions and fewer high-importance tokens. As a result, we retain the top $K$ tokens, where $K$ is determined adaptively based on the Gaussian distribution. Specifically, let $\tau \in [0, 1]$ denotes a threshold of the token elimination, token $x_i$ are kept only if specified as follows:

$$f_{G(z_g)}(c_i(x), c_j(x)) \geq \max_{x \in X}[f_{G(z_g)}(c_i(x), c_j(x))] \times \tau \tag{1}$$

Here, $\tau$ denotes the threshold applied to the TIScores. Additionally, we define a minimum token count $\epsilon$ to ensure a sufficient number of tokens are selected. If the number of tokens satisfying the condition in Equation 1 falls below $\epsilon$, the top-$\epsilon$ tokens are selected based on their TIScores. By setting $\tau$ to a predefined value (e.g., 0.01), Equation 1 enables adaptive token selection, allowing the number of retained tokens to vary according to the size of the IOI. Let $X_s \subseteq X$ represent set of tokens selected under equation 1. The selected tokens are passed through the image encoder, yielding $E(X_s)$, which is then fed into the mask decoder $D(.)$ as shown by the low portion of Figure 4. As noted in Section 3.1, $D(.)$ expects a fixed input size of $H_e \times W_e$. To satisfy this requirement, $E(X_s)$ is zero-padded to the target dimensions, the results is denoted as $Y_s$. The decoder then generates the predicted segmentation mask $M_{pred}$. The following section details the training procedure.

### 3.2.3 TRAINING AND LOSS FUNCTION

Figure 4 illustrates the training and inference workflows of FoSAM, with data flows for inference and training shown in black+blue and black+red, respectively. During inference, only the token $z_g$, which is closest to the gaze location $g$, is used to predict the TIScore. Let $M_{gt}$ represent the ground-truth binary mask corresponding to the input $I$, where $M_{gt,ij} = 1$ indicates that the pixel lies within the IOI, and 0 otherwise. Additionally, let $X_{M_{gt}} \subseteq X$ denote the subset of tokens whose spatial positions fall within the IOI as defined by $M_{gt}$, and let $Z_{M_{gt}} = S(X_{M_{gt}})$ represent the outputs of the $S(.)$ of the corresponding tokens. The loss function used during training comprises three components: the negative log-likelihood, $\mathcal{L}_{\text{nll}}$, the classification loss $\mathcal{L}_{\text{cls}}$, and the Dice loss $\mathcal{L}_{\text{dice}}$. The objective of $\mathcal{L}_{\text{nll}}$ is to train the TIEncoder module $S(\cdot)$ and the Gaussian predictor $G(\cdot)$ to generate a 2D Gaussian map $f_{G(S(x))}(\cdot)$, where $x \in X_{M_{gt}}$, that closely approximates the ground-truth mask $M_{gt}$, as follows:

$$\mathcal{L}_{\text{nll}} = - \sum_{x_m \in X_{M_{gt}}} \sum_{x'_m \in X_{M_{gt}}} log[f_{G(S(x_m))}(c_i(x'_m), c_j(x'_m))] \tag{2}$$

where $\sum_{x'_m \in X_{M_{gt}}} log[f_{G(S(x_m))}(c_i(x'_m), c_j(x'_m))]$ represents the sum of the log-probabilities for each token $x'_m$ within the ground truth mask $M_{gt}$. These probabilities are computed using a 2D Gaussian distribution, where the distribution parameters are generated based on the reference token $x_m$. In other words, we aim to ensure that each token $x_m$ within the ground truth mask $M_{gt}$ can generate a 2D Gaussian distribution that effectively covers the intended IOI.

In practice, obtaining an accurate Gaussian distribution that closely mimic the ground-truth mask poses a granular alignment challenge. For example, when the gaze point lands on the headlight of a car, it is often ambiguous whether the target object is the headlight itself or the entire vehicle. Similarly, if the gaze fixed on a chess piece, it is unclear whether the segmentation should cover the individual piece or the whole chessboard. To mitigate this ambiguity, we introduce an additional classification loss $\mathcal{L}_{\text{cls}}$, which leverages category labels to enforce granular alignment. Beyond alignment, $\mathcal{L}_{\text{cls}}$ also provide clustering effect, markedly enhancing the inter-class diversity of the Gaussian maps. As an auxiliary supervision signal, it encourages the TIEncoder $S(.)$ to learn class-specific features, leading to a better Gaussian maps. Therefore, as shown in Figure 3, a shallow neural network $U(.)$ is attached to the output of $S(.)$ for classification task. Let $M_{cls}(x_i)$ denote the class label associated with the IOI where $x_m$ is mapping to, $\mathcal{L}_{\text{cls}}$ can be expressed as:

$$\mathcal{L}_{\text{cls}} = - \sum_{x_m \in X_{M_{gt}}} CE[U(S(x_m), M_{cls}(x_m)] \tag{3}$$

Here, $CE(\cdot)$ denotes the cross-entropy loss. Finally, to ensure that the output $M_{pred}$ of the image decoder $D(\cdot)$ aligns with the ground-truth mask $M_{gt}$, we apply the Dice loss.

$$\mathcal{L}_{\text{dice}} = DICE[M_{pred}, M_{gt}] \tag{4}$$

The overall loss function can be calculated as $\mathcal{L}_{\text{total}} = \lambda_1 \mathcal{L}_{\text{nll}} + \lambda_2 \mathcal{L}_{\text{cls}} + \lambda_3 \mathcal{L}_{\text{dice}}$, where $\lambda_1, \lambda_2$ and $\lambda_3$ represent the relative importance of the loss functions. During training, only the TIEncoder $S(\cdot)$, classification head $U(\cdot)$, Gaussian head $G(\cdot)$, and image decoder $D(\cdot)$ are updated, while the image encoder $E(\cdot)$ remains frozen, significantly reducing the training cost of FoSAM.

### 3.3 FoSAM STREAMING ALGORITHM

Building on the FoSAM architecture introduced in Section 3.2, this section explains how FoSAM can be extended for efficient instance segmentation across consecutive video frames. As motivated by the study in Section 2.2, FoSAM is triggered during real-time operation only when the input image $I$, captured by the front-facing camera, shows a significant change or when a shift in the user's gaze direction is detected. The logic flow of the FoSAM Streaming Algorithm (FSA) is shown in Figure 5. Let $I_{ini}$ and $I_t$ represent the input images at the initial frame and at time step $t$, respectively. A simple criterion is used to determine whether the current view remains within the

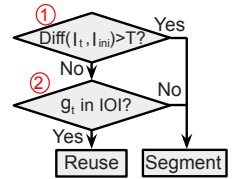

Figure 5: FSA flow.

previous segment by computing the relevant difference between $I_t$ and $I_{ini}$ (Condition 1). Instead of computing pixel-wise differences over the entire images $I_t$ and $I_{ini}$, we restrict the computation to the masked regions, focusing only on the relevant areas across the two frames. Instead of using a fixed

$\alpha$ threshold, we times $\alpha$ by the area of $M_{pred}$. This allows the threshold to adapt automatically based on the size of the gazed object, making the segmentation decision more robust across varying object sizes. If this difference surpasses the threshold $T = \alpha \times \sum_{i,j} M_{pred,ij}$, it signals a substantial change in the previous gazed area, prompting a full re-execution of FoSAM, and $I_{ini}$ will be updated with $I_t$. If no big change is observed, the current gaze location $g_t$ is examined to determine whether it remains in IOI region defined by the segmentation mask $M_{pred}$ of $I_{ini}$ (Condition 2). If it does, $M_{pred}$ can be reused; otherwise FoSAM must be executed with $g_t$ and $M_{pred}$ will be updated accordingly. Details of the FSA are in the supplementary materials.

# 4  EVALUATION

We evaluate FoSAM using publicly available datasets: ADE20K Zhou et al. (2019), LVIS Gupta et al. (2019), and Cityscapes Cordts et al. (2016). As these datasets are originally designed for generic segmentation tasks to cover all the objects in the image, we introduce a gaze-aware masking preprocessing step to enable segmentation over the IOI only. Specifically, for each training and testing sample, a gaze location is randomly selected within the image, and the corresponding IOI region is defined based on this location. The size of the input images is $640 \times 640$ for all datasets.

We use a lightweight two-layer ViT as the TIEncoder $S(.)$. The input images have a resolution of $H \times W = 640 \times$

Table 2: IoU and FLOP comparison of FoSAM and baseline algorithms. K denotes the token budget.

| Method | K | ADE20K | LVIS | Cityscape | GFLOP |
|---|---|---|---|---|---|
| SAM-B Ravi et al. (2024) | 4096 | 0.511 | 0.537 | 0.347 | 744 |
| ESAM-S Xiong et al. (2024) | 4096 | 0.411 | 0.359 | 0.304 | 188.2 |
| ESAM-T Xiong et al. (2024) | 4096 | 0.350 | 0.236 | 0.248 | 56.1 |
| SlimSAM Chen et al. (2024) | 4096 | 0.477 | 0.474 | 0.293 | 15.1 |
| FSNet-DL Zeng et al. (2025) | NA | 0.34 | 0.38 | 0.24 | 18.8 |
| FSNet-SF Zeng et al. (2025) | NA | 0.33 | 0.36 | 0.21 | 12.6 |
| AD-S | 200 | 0.273 | 0.350 | 0.221 | 14.6 |
| LC-S | 200 | 0.368 | 0.477 | 0.385 | 14.6 |
| **FoSAM-S** | 200 | **0.495** | **0.487** | **0.404** | **14.6** |
| AD-S | 100 | 0.252 | 0.343 | 0.203 | 10.4 |
| LC-S | 100 | 0.294 | 0.444 | 0.339 | 10.4 |
| **FoSAM-S** | 100 | **0.465** | **0.461** | **0.362** | **10.4** |
| AD-T | 200 | 0.256 | 0.322 | 0.198 | 7.4 |
| LC-T | 200 | 0.327 | 0.453 | 0.371 | 7.4 |
| **FoSAM-T** | 200 | **0.471** | **0.469** | **0.377** | **7.4** |
| AD-T | 100 | 0.239 | 0.318 | 0.176 | 5.3 |
| LC-T | 100 | 0.264 | 0.421 | 0.302 | 5.3 |
| **FoSAM-T** | 100 | **0.457** | **0.435** | **0.328** | **5.3** |

$640$, and are average downsampled to $H'_e \times W'_e = 160 \times 160$ prior to processing. The FoSAM framework is integrated with ESAM Small Xiong et al. (2024) and ESAM Tiny, referred to as FoSAM-S and FoSAM-T, respectively. Two baseline methods are developed for comparison. The first method, *Average Downsampling* (AD), enhances segmentation efficiency by directly applying average pooling to the original input $X$, reducing its spatial resolution and thus lowering computational cost. The second method, *Local Cropping (LC)*, improves efficiency by extracting a fixed-size patch centered around the gaze point $g$, thereby narrowing the focus to a smaller input region. These baselines are evaluated on both ESAM Small and ESAM Tiny, resulting in AD-S, AD-T, LC-S, and LC-T configurations. Furthermore, we evaluate performance against SAM Base (SAM-B), ESAM Small (ESAM-S), ESAM Tiny (ESAM-T), and SlimSAM Chen et al. (2024), which improves segmentation efficiency through a combination of knowledge distillation, embedding pruning, and bottleneck pruning. We also compare FoSAM with FSNet, which integrates DeepLab He et al. (2016) and Segformer Xie et al. (2021) backbones to enable efficient segmentation via learnable input downsampling strategies, referred to as FSNet-DL and FSNet-SF, respectively. During training, the weights for the loss terms $\mathcal{L}_{nll}$, $\mathcal{L}_{cls}$, and $\mathcal{L}_{dice}$ are all set to 1. We evaluate segmentation performance using the Intersection over Union (IoU) metric, which measures the overlap between the predicted segmentation and the ground truth IOI region. To control the average number of retained tokens $K$, we adjust the TIScore threshold $\tau$ and the minimum token count $\epsilon$ as defined in Equation 1. More evaluation results are shown in the supplementary materials.

## 4.1  EVALUATION RESULTS OF FOSAM

Table 2 presents the evaluation results. The compared methods, FoSAM, AD, and LC, all operate on input images at their original resolution. FoSAM adopts an adaptive token sampling strategy with average token budgets of $K = 200$ and $K = 100$ across the entire test dataset. To ensure a fair comparison, the input sizes of AD and LC are adjusted so that their computational cost aligns with that of FoSAM. For ESAM, SlimSAM, and SAM, regardless of the original input size, the built-in

Table 3: Processing latencies of FoSAM on an AR platform.

| Latency (ms) | Token 200 | 100 |
|---|---|---|
| FoSAM-S | 21.6 | 18.3 |
| FoSAM-T | 17.6 | 14.5 |

Figure 6: Average IoU and frame skip rate of FSA on FoSAM-S and FoSAM-T under varying $\alpha$.

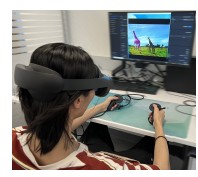

Figure 7: Participants are doing user study.

resizing layer by default resizes all inputs to $1024 \times 1024$, resulting in a fixed number of 4096 tokens. For SlimSAM, we tune its pruning ratio to achieve a similar computational cost to FoSAM. In the case of FSNet-DL and FSNet-SF, input images are downsampled to a resolution of $80 \times 80$, which is the default setting of it. As shown in the results, FoSAM consistently outperforms the AD and LC baselines across all datasets. In contrast to different versions of ESAM, SlimSAM and FSNet, FoSAM achieves both higher accuracy and reduced computation. For instance, in Cityscape datasets, FoSAM-T with 100 token budget achieves $3.5\%$ higher IoU than SlimSAM with only one third of the computation cost. This highlights the effectiveness of IOI guided adaptive token sampling in FoSAM. Moreover, FoSAM demonstrates strong generalizability across datasets, with minor performance variations attributed to dataset-specific characteristics. Compared to SAM, although FoSAM does not achieve equally high IoU, it provides comparable accuracy while reducing computational cost by over $50\times$. As shown in the user study in Section 4.4, FoSAM also offers a significantly better visual experience, whereas SAM introduces noticeable processing delays that degrade user experience.

To assess FoSAM's processing latency on AR device, we profile the performance of FoSAM-S and FoSAM-T with average token budgets of K=200 and 100 on the Qualcomm XR2 Gen2 platform Qualcomm Technologies, Inc. (2024b), utilizing the Qualcomm AI Hub toolkit Qualcomm Technologies, Inc. (2024a). It is important to note that the Qualcomm XR2 Gen2 is integrated into the Meta Quest Pro for AR/VR task processing Inc. (2022); Engel et al. (2023). The average latencies, computed over 100 profiling runs on a single LVIS sample, are reported in Table 3. Compared to the latency of SAM and ESAM shown in Table 1, FoSAM achieves an average of $24\times$ reduction in inference time and meets the 30 ms latency threshold required for a seamless visual experience, as indicated by prior research Kaaresoja et al. (2014); vis; Albert et al. (2017).

## 4.2 EVALUATION RESULTS OF FoSAM STREAMING ALGORITHM

The FSA mechanism introduced in Section 3.3 reduces segmentation overhead by reusing results from previous frames. However, as illustrated in Figure 5, varying the parameter can influence the average IoU across frames. To evaluate this effect, we analyze the performance of FoSAM-S and FoSAM-T with FSA evaluated on the Cityscapes dataset under different settings of $\alpha$, which represent the threshold for image difference shown in Figure 5. The token budget is set to 200 for both FoSAM-S and FoSAM-T. We use 10 video sequences which contain 5000 images in total. However, since the Cityscapes dataset lacks gaze location data, we incorporate gaze traces described in Lv et al. (2024) into each frame of the Cityscapes dataset. The results are described in Figure 6, as $\alpha$ increases from 0.01 to 0.1, more frames are skipped by reusing segmentation results of previous frames. For instance, skipping $40\%$ of the frames leads to only a 0.03 reduction in average IoU across frames for both FoSAM-S and FoSAM-T, showing that FSA can greatly reduce segmentation computation with negligible impact on the accuracy.

## 4.3 ABLATION STUDY

**Impact of TIEncoder Size** In this section, we examine how the computational cost of the TIEncoder $S(\cdot)$ influences FoSAM's accuracy under an average token budget of $K = 200$ on the ADE20K dataset. Specifically, we evaluate the impact on FoSAM performance by varying both the parameter size of $S(\cdot)$ and the input resolution of $X'$ fed into $S(\cdot)$. As shown in Table 4, increasing the number of encoder layers from 2 to 3 and raising the input resolution from $160 \times 160$ to $640 \times 640$ leads to only a

Table 4: Impact of the TIEncoder.

| Layer Number | Image Size | IoU | SE GFLOPs |
|---|---|---|---|
| 3 | $640 \times 640$ | 0.499 | 10.4 |
| 3 | $320 \times 320$ | 0.499 | 2.8 |
| 2 | $320 \times 320$ | 0.497 | 1.88 |
| 2 | $160 \times 160$ | 0.495 | 0.48 |
| 2 | $80 \times 80$ | 0.492 | 0.12 |
| 1 | $320 \times 320$ | 0.492 | 0.94 |

slight improvement in IoU, while the computational cost rises substantially from 0.48 GFLOPs to 10.4 GFLOPs for the TIEncoder processing. Additionally, reducing the number of encoder layers results in a more significant accuracy drop than downsampling the input image. The TIEncoder design in FoSAM strikes an effective balance between computational efficiency and performance.

**Impact of the Token Budget**   As described in Section 3.2.2, FoSAM adopts an adaptive token sampling mechanism that adjusts the number of retained tokens according to the size of the IOI. In contrast, the LC algorithm applies a fixed token count regardless of IOI size, which can lead to too few tokens for large IOIs, reducing accuracy, or too many tokens for small IOIs, causing unnecessary computation. Table 5 compares the performance of FoSAM and LC under varying token budgets $K$ on different datasets. For FoSAM, the number of retained tokens varies across frames, with an average of $K$. On the LVIS, FoSAM-S achieves an average of 0.9% higher IoU than LC-S when $K = 400$, and outperforms LC-S by 5.5% when $K = 25$. These results highlight FoSAM's ability to adapt the token budget to the size of the IOI, allowing it to maintain accuracy even under constrained token budgets.

Table 5: Performances with varying token budgets.

| Dataset | Budget (K) | FoSAM-S | LC-S |
|---|---|---|---|
| LVIS | 400 | 0.499 | 0.490 |
| | 200 | 0.487 | 0.477 |
| | 100 | 0.475 | 0.443 |
| | 25 | 0.371 | 0.318 |
| Cityscape | 400 | 0.417 | 0.394 |
| | 200 | 0.404 | 0.385 |
| | 100 | 0.362 | 0.339 |
| | 25 | 0.269 | 0.243 |
| ADE | 400 | 0.517 | 0.442 |
| | 200 | 0.499 | 0.368 |
| | 100 | 0.472 | 0.293 |
| | 25 | 0.311 | 0.227 |

**TIEncoder design: Gaussian vs. Token-wise**   Selecting tokens based on assigned scores is a common concept in computer vision, but most of them use a token-wise design, which means predicting a score for each token separately, as described in Section 3.2.1. In contrast, FoSAM only predict a single Gaussian distribution and use it to assign

Table 6: Gaussian vs. Token-wise

| Method | Lvis (200 token budget) (IoU) |
|---|---|
| Gaussian (ours) | 0.487 |
| Token-wise design | 0.231 |

all TIScores at once. Table 6 provides a comparison to demonstrate the effect of switching to a common token-wise design in our case by adding Importance Head for each token. As shown, the token-wise approach performs significantly worse under the same token budget. It is because unlike the smooth and concentrated Gaussian TIScore map, the token-wise design often produces fragmented and discontinuous token selection, and the SAM's mask decoder is very vulnerable to this discontinuity.

### 4.4   USER STUDY

To assess the user experience benefits of FoSAM over the standard SAM method on AR device, we conduct a two-interval forced-choice (2IFC) Yeshurun et al. (2008) user study simulating both methods' effects on the Meta Quest Pro headset Inc. (2022). Segmentation masks were precomputed and gaze-contingent visualizations were rendered with artificially introduced latencies reflecting each method's runtime profiled on the Qualcomm XR2 Gen2's NPU Qualcomm Technologies (2025). Each of the seven participants compared results across four test images, selecting the preferred output in 32 trials, as shown in Figure 7. As shown in Figure 8, FoSAM was preferred in 96.9%±4.8% of trials overall, consistently outperforming SAM

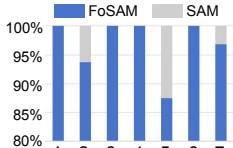

Figure 8: Preference rates of the seven participants.

across all images. Although FoSAM incurs a slight drop in segmentation accuracy compared to SAM, its greatly lower latency and better temporal alignment with gaze location result in a significantly improved user experience. See supplementary materials for implementation details.

## 5   CONCLUSION AND LIMITATION

We presented FoSAM, a gaze-guided segmentation framework optimized for AR/VR. By focusing computation on user-relevant regions, FoSAM achieves over $50\times$ speedup with minimal accuracy loss. Experiments and user studies validate its effectiveness for real-time deployment on AR devices. While FSA achieves substantial latency reduction, it relies on heuristic thresholds for result reuse, which could be further optimized in future work. Additionally, potential risks include privacy concerns related to eye-tracking data and reliance on model accuracy in sensitive applications, which needs to be solved as future work.

**Ethics Statement**    We affirm that all authors have read and will adhere to the ICLR Code of Ethics. Our work includes a user study with seven adult participants using a head-mounted AR devices to compare gaze-contingent segmentation latency/quality. All participants provided informed consent prior to participation. The study involved minimal risk, and participants could withdraw at any time without penalty. No personally identifiable information (PII) was collected; no raw eye images were stored; only anonymized preference choices and aggregate metrics were retained. The study complied with our institution's human-research guidelines and applicable laws. We use only public datasets for training/evaluation. All data are used under their respective licenses/terms; no attempt was made to re-identify individuals or to reconstruct any personally sensitive attributes. We will release code, configuration files, and evaluation scripts sufficient for reproducibility. We will not release any user-study raw data or raw eye-tracking images.

**Reproducibility Statement**    We aim to make all results fully reproducible. We provide an anonymized repository at `https://anonymous.4open.science/r/FoSAM-D627` which contains the full source code. The model architecture and inference pipeline are specified in Section 3 (with the streaming procedure detailed in Appendix A.1), and the learning objective is given by Equation 2,3,4. Detailed Hyperparameter settings for experiments can be find in the provided anonymized repository. Datasets preprocessing are described in Section 4.

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

# A APPENDIX

## A.1 DETAILS OF FOSAM STREAMING ALGORITHM

FoSAM streaming algorithm is outlined in Algorithm 1, a simple criterion is applied to detect if the view is still in the previous video fragment (VF) by calculating the relative difference between $I_t$ and $I_{ini}$ (line 4). If this difference exceeds a threshold $T = \alpha \times \sum_{i,j} M_{pred,ij}$, it indicates a significant change in the front scene, triggering a full re-execution of FoSAM (line 5). $I_{ini}$ and $M_{pred}$ will be updated accordingly (line 6). If not, the current gaze location is analyzed to determine if it remains within the same IOI region (line 9). If it does, the last segmentation mask $M_{pred}$ can be reused (line 10). Otherwise, FoSAM must be executed with the new input frame $I_t$ and gaze location $g_t$ (line 12), $I_{ini}$ and $M_{pred}$ will be updated accordingly (line 13).

---

**Algorithm 1:** FoSAM Streaming Algorithm

**Input:** $I_{ini}$ is the initial frame of current VF. $M_{pred}$ is the buffered segmentation mask of $I_{ini}$. IOI is the instance of interest region defined by $M_{pred}$. $g_t$, $I_t$, and $M_t$ represent the current gaze direction, the current input frame, and the segmentation mask at time $t$. $\alpha$ is the threshold of result reuse.

1 **Initiation**
2    $I_{ini} = \varnothing$, $M_{pred} = \varnothing$
3    **for** $1 \leq t \leq T$ **do**
4      **if** $\frac{\sum_{ij} |I_{t,ij} - I_{ini,ij}|}{\sum_{ij} I_{ini,ij}} > \alpha \times \sum_{i,j} M_{pred,ij}$ **then**
5        Run FoSAM with $I_t$ and $g_t$, get $M_t$;
6        $I_{ini} \leftarrow I_t$, $M_{pred} \leftarrow M_t$;
7        **return** $M_t$
8      **else**
9        **if** $g_t$ *in IOI* **then**
10          **return** $M_{pred}$
11        **else**
12          Run FoSAM with $I_t$ and $g_t$, get $M_t$;
13          $I_{ini} \leftarrow I_t$, $M_{pred} \leftarrow M_t$;
14          **return** $M_t$

---

## A.2 EXTRA EVALUATION

**Noise and latency sensitivity study**
Relevant studies show that gaze jitter keeps most of the gaze points within $1°$ of visual angle from the target center. Including the eyetracker's error, the overall gaze error remains within $2°$. Therefore, We add random angular noise into the gaze location and report the performance under the same token budget. As shown in Table 7, on LVIS (with many small objects), slight gaze noise can push fixation outside the target boundary and modestly reduce accuracy. On ADE, FoSAM remains robust.

A typical eye tracker's refresh rate is 100 Hz, introducing 10ms of gaze latency. Therefore, we use the gaze point from 10 ms earlier as the current frame's gaze to simulate this delay, and report its impact on Aria with the same token budget. As shown in Table 8, the accuracy degradation from the gaze latency is minimal, highlighting the robustness of FoSAM to gaze latency.

Table 7: Impact of the Gaze Noise.

| Gaze error (deg) | 0 (no error) | ±1 | ±2 |
|---|---|---|---|
| Lvis (200 tokens budget) (IoU) | 0.487 | 0.461 | 0.437 |
| ADE (200 tokens budget) (IoU) | 0.495 | 0.488 | 0.479 |

Table 8: Impact of the Latency.

| Gaze latency | negligible latency | 10ms |
|---|---|---|
| AriaEveryday (400 token budget) (IoU) | 0.588 | 0.564 |

As described in Section 3.2.3, during training we supervise all tokens within the ground-truth mask to produce the same target Gaussian. This means any fixation locations within the object are encouraged to yield the same target Gaussian TIScore map. This training technique (like data augmentation) ensures that as long as the gaze remains on the object, FoSAM produces consistent predictions.

**Impact of Adaptive Token Sampling Hyperparameters** The saliency score threshold $\tau$ and the minimum token count $\epsilon$ are used to determine how many tokens are retained, as detailed in Section 3.2.2. Different combinations of $\tau$ and $\epsilon$ can result in the same average number of tokens being retained across the samples in a dataset. In Table 9, we evaluate the effects of varying $\tau$ and $\epsilon$ under different token budgets using FoSAM-S over the Cityscape dataset. A larger $\tau$ reduces the number of tokens selected based on saliency scores; to maintain the target token budget, $\epsilon$ must be increased accordingly.

When the Token Budget is held constant—*i.e.*, the mean of $|X_s|$ is fixed—a smaller $\tau$ and $\epsilon$ result in a larger variance of $|X_s|$ across different samples. In this setting, $X_s$ most faithfully reflects the Gaussian distribution predicted by the model. However, our experiments reveal that FoSAM may produce overly extreme Gaussian distribution, especially when the predicted Gaussian has a very small standard deviation, overall performance deteriorates markedly. As shown in Table 9, a smaller $\epsilon$ indeed causes a slight drop in performance, underscoring the necessity of introducing the $\epsilon$ term. Conversely, larger $\tau$ and $\epsilon$ prevent $X_s$ from accurately reflecting predicted Gaussian distribution, constraining FoSAM's capacity to allocate token budgets in proportion to IOI size and resulting in a pronounced performance degradation.

| $\tau$ | $\epsilon$ | FoSAM-S | Token Budget (K) |
|---|---|---|---|
| 0.2 | 300 | 0.404 | 400 |
| 0.12 | 200 | 0.417 | 400 |
| 0.1 | 100 | 0.416 | 400 |
| 0.4 | 150 | 0.39 | 200 |
| 0.3 | 100 | 0.404 | 200 |
| 0.25 | 50 | 0.402 | 200 |
| 0.7 | 75 | 0.355 | 100 |
| 0.6 | 50 | 0.362 | 100 |
| 0.55 | 25 | 0.362 | 100 |

Table 9: Evaluation results of adaptive token sampling settings.

**Impact of Loss Weights**    During the training of FoSAM, the overall loss is computed as a weighted sum of the negative log-likelihood loss, classification loss, and Dice loss, with weights $\lambda_1$, $\lambda_2$, and $\lambda_3$, respectively, as defined in Equation 2, Equation 3, and Equation 4. Dice loss plays a critical role in the segmentation task, while the negative log-likelihood loss and classification loss help the saliency encoder in FoSAM better adapt the saliency score distribution based on the IOI. In Table 10, we evaluate the contributions of the negative log-likelihood loss and the classification loss to the overall performance of FoSAM by varying their weights $\lambda_1$ and $\lambda_2$, using the FoSAM-S model on the Cityscapes dataset.

| $\lambda_1$ | $\lambda_2$ | Token Budget | IoU |
|---|---|---|---|
| 1 | 1 | 200 | 0.404 |
| 2 | 1 | 200 | 0.398 |
| 1 | 0 | 200 | 0.381 |

Table 10: Impact of the weights of the nll loss ($\lambda_1$) and classification loss ($\lambda_2$).

The setting $\lambda_1 = 1, \lambda_2 = 1$ is used as the baseline for our main evaluation results. Increasing $\lambda_1$ reduces the relative contribution of the classification loss, which leads to a decrease in segmentation accuracy. When $\lambda_2 = 0$, the classification loss is completely removed, resulting in a further drop in accuracy. This highlights the importance of the classification loss in achieving good segmentation performance.

### A.3    DETAILS OF USER STUDY

To evaluate the enhancement in user experience offered by FoSAM compared to the conventional SAM approach, we simulate their visual effects on the Meta Quest Pro HMD Inc. (2022) due to HMD's restrictions on direct access to its NPU Qualcomm Technologies (2025).

For each test image, segmentation masks for the IOIs were precomputed using FoSAM-S and

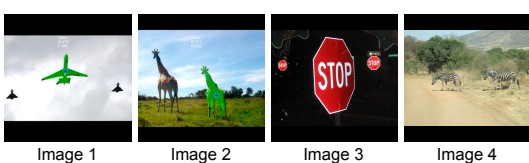

| Image 1 | Image 2 | Image 3 | Image 4 |

Figure 9: Four sample images.

SAM-B Kirillov et al. (2023). During the user study, the HMD displays the mask of the currently observed IOI according to gaze locations obtained by the HMD's gaze tracker, simulating the visual effects of both algorithms.

To account for system performance, we profile the latency of FoSAM-S (200 token budget) and SAM-B on the Qualcomm XR2 Gen2 platform Qualcomm Technologies, Inc. (2024b) using the Qualcomm AI Hub toolkit Qualcomm Technologies, Inc. (2024a), and artificially introduce the delay between the moment the user's gaze identifies the IOI and when the segmentation mask is displayed on the VR screen. Due to its efficient design, FoSAM-S exhibited significantly lower latency (21.6 ms) compared to SAM-B (1897.4 ms). Figure 9 illustrates these differences: in Image 1, FoSAM maintains low latency, ensuring masks closely align with current gaze location, whereas in Image 2, SAM's higher latency results in noticeable misalignment between the mask and current gaze location, negatively impacting user experience.

Seven participants take part in the study (in Figure 7, where the computer monitor display the HMD-cast content), and interact using the HMD controllers. The stimuli consists of four images (Figure 9). The two methods, denoted as $m_1$ (FoSAM-S) and $m_2$ (SAM-B), are directly compared. Participants

perform a two-interval forced-choice (2IFC) task Yeshurun et al. (2008), viewing two segmentation results ($t_1$ and $t_2$) per image, with masks and latency applied as described. The test conditions ($t_1$ and $t_2$) are randomly assigned to $m_1$ and $m_2$. Using the HMD controller buttons, participants switch between $t_1$ and $t_2$ while observing different objects. After comparing both at least once, they select the one with higher perceived quality. Each participant completes 32 trials, consisting of 4 images, each tested with both $t$ and $m$ pairs across 4 repetitions, presented in a random sequence.

Figure 8 presents the results. Across participants, FoSAM was preferred in 96.9%±4.8% of trials overall, consistently outperforming SAM across all images (98.2%±4.7% for Image 1, 96.4%±6.1% for Image 2, 96.4%±9.4% for Image 3, and 96.4%±9.4% for Image 4). These results evidence that the superior performance of FoSAM can potentially improve user experience for AR/VR applications due to lower processing latency and precise segmentation accuracy.

### A.4 USE OF LARGE LANGUAGE MODELS (LLMs)

In accordance with the ICLR Author Guide, we disclose that we used LLM-based assistants only for English-language editing (grammar, wording, and minor rewrites for clarity/flow). LLMs did not generate ideas, methods, analyses, figures/tables, code, experiments, or results.

