# OpenReview forum: "FoSAM: Focus-Oriented Adaptive Token Sampling for Efficient Segment Anything in Augmented Reality"
_ICLR.cc/2026/Conference — ICLR 2026 Conference Withdrawn Submission_

### Official Review · Reviewer_GshQ · 2025-10-22

**Soundness:** 3
**Presentation:** 3
**Contribution:** 2
**Rating:** 4
**Confidence:** 4

**Summary:**

The paper proposes FoSAM, a gaze-guided segmentation framework that reduces computation by focusing on user-attended regions. Built on ESAM, it utilizes a Gaussian predictor to model the shape and extent of the gaze area, selectively processing only the most relevant tokens. A simple frame-reuse mechanism is also introduced to avoid redundant computation across consecutive frames. The overall idea is intuitive and well-motivated, aiming to make segmentation more efficient for AR/VR devices.

**Strengths:**

1. The paper addresses a relevant problem of improving segmentation efficiency for AR/VR applications. The motivation and problem setup are clear, and the proposed method is easy to follow.

2. The approach considers both spatial efficiency through adaptive token selection and temporal efficiency through frame reuse, making the overall framework coherent.

3. The experiments are comprehensive and generally support the main claims. The authors also provide code and detailed implementation information, which makes the work reproducible.

**Weaknesses:**

1. The work has limited novelty, as the use of gaze or foveated information for efficient segmentation is not new, and the method mainly extends existing frameworks.

2. Some design components are overly simplistic, such as the frame-change detection, which merely measures the percentage of absolute pixel difference between frames.

3. The paper lacks systematic hyperparameter tuning, making it unclear whether the reported performance is optimal.

**Questions:**

1. While latency is clearly critical for AR applications, the paper could better motivate why continuous segmentation is necessary, rather than relying on occasional or event-triggered updates.

2. Segmentation often benefits from contextual information. It would be helpful to clarify how FoSAM maintains accuracy, especially when the gaze region is smaller than the target object or lacks sufficient context.

3. The method applies a hard token selection strategy. Have the authors considered softer or hierarchical compression mechanisms that might retain context more smoothly?

4. Since FoSAM depends on external gaze-tracking input, it would be interesting to discuss how the approach generalizes when such input is noisy or unavailable, or whether learning-based gaze estimation could be incorporated.

5. Relatedly, does the model support or could it be extended to handle multiple or smoothed gaze points to improve temporal stability?

6. The frame-change detection criterion appears rather simple, based mainly on pixel-level difference ratios. More details on how segmentation masks are reused or mapped between frames would strengthen the technical clarity.

7. The paper does not describe any grid search or tuning process for key hyperparameters. Clarifying whether such tuning was attempted would help establish fairness and reproducibility.

---

### Official Review · Reviewer_6BSe · 2025-10-30

**Soundness:** 4
**Presentation:** 3
**Contribution:** 3
**Rating:** 4
**Confidence:** 4

**Summary:**

This paper proposes FoSAM, a focus-oriented adaptation of the Segment Anything Model (SAM) for augmented reality (AR) applications. The key idea is to integrate real-time gaze tracking to prioritize segmentation only in the user’s region of interest (ROI). This allows the system to avoid running full-frame segmentation on every frame, reducing computation by 50× while maintaining alignment quality of AR overlays. A user study is included to evaluate real-world performance, and code is provided for reproducibility.

**Strengths:**

1. Clear Practical Relevance. Image segmentation is one of the computational bottlenecks in AR systems, especially on-device. Linking segmentation priority to gaze attention is well-motivated and directly tied to human visual processing.
2. SAM is powerful but expensive. The paper smartly adapts SAM instead of proposing yet another segmentation model.
3. Using gaze prioritization aligns system computation with user perceptual importance, a strong conceptual basis.

**Weaknesses:**

1. It is unclear how the gaze coordinates are converted into segmentation prompts.
2. Reducing computation may lead to reduced segmentation accuracy on non-attended areas.
3. A 50× reduction is impressive, but the baseline must be clearly defined

**Questions:**

Are comparisons done on-device or on desktop hardware?

---

### Official Review · Reviewer_apYi · 2025-10-31

**Soundness:** 3
**Presentation:** 3
**Contribution:** 1
**Rating:** 2
**Confidence:** 4

**Summary:**

This paper presents FoSAM (Focus-Oriented Segment Anything Model), a gaze-guided adaptation of SAM aimed at efficient, real-time segmentation for AR/VR applications. FoSAM employs eye-tracking data to focus segmentation on the user’s gaze region through a Gaussian-guided adaptive token sampling strategy, significantly reducing computational cost. Additionally, the FoSAM Streaming Algorithm (FSA) enables temporal reuse of segmentation masks across consecutive frames when gaze and scene remain stable. Experiments on ADE20K, LVIS, and Cityscapes demonstrate over 50× computational speed-up while maintaining comparable segmentation accuracy. Profiling on the Qualcomm XR2 platform shows real-time inference (<30 ms latency). A small user study using the Meta Quest Pro suggests that FoSAM offers a smoother and more natural visual experience. Overall, the work provides a well-engineered, perception-aware optimization of SAM for AR devices, focusing more on practical deployment and user experience than on algorithmic novelty.

**Strengths:**

The work addresses an important engineering problem—making large-scale segmentation models feasible for resource-limited AR headsets. The idea of aligning computational attention with human gaze is intuitive and well motivated by foveated rendering principles. Integrating this idea into EfficientSAM through token-level adaptation is technically sound and effectively implemented. The computational efficiency gains (over 50× speed-up) are impressive and thoroughly validated through quantitative benchmarks. The addition of the streaming algorithm (FSA) for frame reuse further demonstrates thoughtful design for real-time performance. The user study, though small, adds qualitative evidence that the proposed method improves visual smoothness and subjective comfort. The writing is clear, the experimental design is detailed, and the implementation seems reproducible. Overall, the paper stands out as a solid systems contribution bridging computer vision and AR usability, demonstrating a concrete pathway to deploy segmentation models efficiently in real-world, interactive settings.

**Weaknesses:**

The main limitation is the lack of methodological novelty. While the combination of gaze-guided processing and adaptive token selection is well executed, each component is derived from existing concepts such as EfficientSAM, Gaussian saliency, and token sparsification. The paper lacks a new learning paradigm or theoretical formulation. From an ICLR standpoint, which prioritizes representational and algorithmic contributions, this makes the work less competitive. The Gaussian-based token scoring remains heuristic, without learning-driven adaptation or formal justification. Moreover, the user study evaluates only perceptual preference (latency and visual comfort), not task-based performance or generalization to other AR applications. The method is also highly specialized to eye-tracked AR devices, limiting broader relevance. Finally, there is little analysis of how gaze-guided token selection affects learned representations or attention maps—insight that would have made it more meaningful for the ML community. Overall, a strong engineering paper but limited in research novelty.

**Questions:**

In addition to the weakness, it would be great if authors can response to the following minor comments.
- The mathematical derivation of the Gaussian importance model (Eq. 1–4) could be clarified through diagrams or simplified notation.
- Since the paper focuses on AR/VR, a supplementary video demonstration is strongly recommended. Visual evidence of real-time segmentation, latency comparison with SAM, and gaze tracking would substantially improve the reader’s understanding and the credibility of the real-time claims.

**Details Of Ethics Concerns:**

Although the authors state that the user study complied with their institution’s human-research guidelines, the experiment clearly involves human participants interacting with AR systems and eye-tracking data. Therefore, it would be appropriate — and in line with common research ethics standards — for such a study to undergo formal Institutional Review Board (IRB) or equivalent ethical review.
Even if the study poses minimal risk, obtaining explicit IRB approval (or stating that an exemption was granted) would strengthen the paper’s ethical transparency and ensure compliance with best practices for research involving human subjects.

---

### Official Review · Reviewer_Qw6L · 2025-11-01

**Soundness:** 2
**Presentation:** 2
**Contribution:** 2
**Rating:** 4
**Confidence:** 4

**Summary:**

This paper introduces FoSAM, a framework designed to reduce the computational cost of segmentation tasks by incorporating user gaze direction. In brief, instead of processing a full sequence of tokens representing the entire image, the proposed pipeline is gaze-guided and focuses only on the tokens corresponding to the user’s viewing region. This approach eliminates redundant tokens that represent non-essential areas.
The motivation is supported by a user study showing that within a video frame segment, the user’s gaze typically changes minimally. Therefore, it is reasonable to reduce the number of tokens based on gaze direction and to reuse segmentation results when gaze shifts are minor.
In the proposed pipeline, a lightweight TIEncoder is trained to produce a gaze-guided mask to indicate exactly which tokens should be focused. This mask is then used by the sampling module to select a reduced set of tokens, which are subsequently fed into the pre-trained ESAM model to generate the final segmentation map. This process significantly lowers computational cost compared to the original model.

**Strengths:**

+ The motivation is clear and reasonable. The user behavior study is carefully conducted and strongly supports the authors’ claim that focusing only on the gaze point is a valid approach to reducing computational cost.
+ The computational cost is indeed reduced by approximately 50 times compared to ESAM while achieving higher performance, demonstrating the effectiveness of gaze-guided efficient segmentation as claimed by the authors.
+ The evaluations are well designed and conducted for both single-frame and video scenarios.

**Weaknesses:**

- This paper benefits from a carefully conducted user study, which supports the claim that focusing only on the IOI is reasonable for achieving lower computational cost. However, the proposed method lacks novelty. If the gaze mask is considered as auxiliary information, there are already many similar studies that perform auxiliary information-guided segmentation, which are not discussed in this work, such as [a] and [b].
 [a] Object-guided instance segmentation with auxiliary feature refinement for biological images. IEEE Transactions on Medical Imaging (TMI).
 [b] Guided Filter Network for Semantic Image Segmentation. IEEE Transactions on Image Processing (TIP).
- Although this study reduces the computational cost (GFLOPs) by sampling a sequence of tokens down to K tokens through a sampling mechanism based on the predicted Gaussian gaze mask, it seems that the model size increases due to the introduction of TIEncoder (S), Gaussian Encoder (G), and Decoder (D). The architecture of G is not described. Therefore, in scenarios where the model is integrated into edge devices such as AR glasses, the increased model size could be a challenge.
- The evaluation is somewhat not comprehensive. Based on the current description, it appears that only one random gaze mask is provided for each image to simulate the user’s gaze direction. To support the claim, several gaze masks should be generated for each image to evaluate the diversity of user attention.
- The evaluation of FoSAM streaming is conducted only using FoSAM itself. If FoSAM streaming is intended to be a pluggable processing step (as shown in Fig. 5), it could also be adapted to FSNet as a comparative baseline, which would make the comparison more convincing.

**Questions:**

1.	In the comparisons presented in Table 2, are all comparative baselines trained and tested using the same generated gaze masks? If the gaze masks are randomly generated for each baseline, the comparison could be considered unfair, as random gaze masks may affect performance. If the same gaze masks are consistently used across all baselines, this information should be clearly stated.
2.	The architecture of the Gaussian predictor G and the total model size comparisons for all baselines in Table 2 should be provided.
3.	Does the reported GFLOPs of FoSAM include the Gaussian generator and sampling processes? Methods such as AD and LC seem not to include these processes, yet they have similar GFLOPs to FoSAM.
4.	Since comparative baselines such as SAM and E-SAM appear to be only pre-trained and not fine-tuned, while FoSAM is fine-tuned for the Gaussian predictor, it would be helpful to provide a discussion on the trade-off between efficiency and additional training time. Specifically, the authors could include the training cost (e.g., training time and GFLOPs incurred during training).

**Details Of Ethics Concerns:**

N.A.

---

### Note · Authors · 2025-11-14

I have read and agree with the venue's withdrawal policy on behalf of myself and my co-authors.